# Understanding Bacteriophage Tail Fiber Interaction with Host Surface Receptor: The Key “Blueprint” for Reprogramming Phage Host Range

**DOI:** 10.3390/ijms232012146

**Published:** 2022-10-12

**Authors:** Jarin Taslem Mourosi, Ayobami Awe, Wenzheng Guo, Himanshu Batra, Harrish Ganesh, Xiaorong Wu, Jingen Zhu

**Affiliations:** 1Bacteriophage Medical Research Center, Department of Biology, The Catholic University of America, Washington, DC 20064, USA; 2Program in Cellular and Molecular Medicine, Boston Children’s Hospital, Harvard Medical School, Boston, MA 02115, USA; 3VCU Life Sciences, Virginia Commonwealth University, Richmond, VA 23284, USA

**Keywords:** bacteriophage (phage), T4 phage, tail fiber, tail fiber structure, tail fiber engineering, phage–host interaction, phage host range, machine learning

## Abstract

Bacteriophages (phages), as natural antibacterial agents, are being rediscovered because of the growing threat of multi- and pan-drug-resistant bacterial pathogens globally. However, with an estimated 10^31^ phages on the planet, finding the right phage to recognize a specific bacterial host is like looking for a needle in a trillion haystacks. The host range of a phage is primarily determined by phage tail fibers (or spikes), which initially mediate reversible and specific recognition and adsorption by susceptible bacteria. Recent significant advances at single-molecule and atomic levels have begun to unravel the structural organization of tail fibers and underlying mechanisms of phage–host interactions. Here, we discuss the molecular mechanisms and models of the tail fibers of the well-characterized T4 phage’s interaction with host surface receptors. Structure–function knowledge of tail fibers will pave the way for reprogramming phage host range and will bring future benefits through more-effective phage therapy in medicine. Furthermore, the design strategies of tail fiber engineering are briefly summarized, including machine-learning-assisted engineering inspired by the increasingly enormous amount of phage genetic information.

## 1. Introduction

Bacteriophages (phages) are widely distributed on land and seas, including extreme environments, and probable constitute the largest biomass in the biosphere [1,2,3]. Most phages package their genome in the proteinaceous capsid (or head) and have a tail attached to the capsid [4]. Tailed double-stranded DNA bacteriophages belonging to the class *Caudoviricetes* (*Cauda* means “tail” in Latin) are the most prevalent (~96% of all known phages) [5,6]. Based on tail morphology, they are further classified into three morphotypes: myovirus, siphovirus, and podovirus [7]. Myophages (e.g., T4, T2, Mu, S16, and φKZ) have long, rigid, contractile tails with a sheath around a central tube; siphophages (e.g., λ, T5, HK97, and SPP1) possess long, flexible, non-contractile tails; and podophages (e.g., T7, T3, P22, and φ29) have short, non-contractile tails. Of these, myophages possess the most complex tail architectures with the greatest number of proteins involved in tail assembly and function [8].

The phage tail is the specialized nano-machinery that specifically recognizes bacterial host cells, penetrates the cell wall and/or membrane, and ejects the phage genome into the host cytosol to produce new viral particles [7]. The tail fibers (or spikes), located at the distal end of the tail, mediate phage binding to a specific receptor present on the cognate bacterial host surface, such as lipopolysaccharide (LPS), porin transmembrane proteins, teichoic acids, and even organelles (e.g., pili or flagella) [9,10,11]. The tail fibers (or spikes) primarily determine the host specificity (or range) and phage infection process [12,13,14,15]. The diverse repertoire of phage tail fibers (or spikes) [16,17,18] ensures effective recognition and adsorption by a wide variety of bacterial hosts, resulting in phages as the most abundant and diverse biological entities on Earth [2,16,19].

In contrast to antibiotics, which effective target a broad spectrum of bacterial strains, most phages infect only a very limited range of hosts because host recognition occurs via a specific interaction of the phage tail fibers with the host receptors. This high specificity is considered an advantage, as phages kill the targeted bacterial pathogens without harming natural microorganisms (phage therapy) [20,21]. However, it is also a distinct disadvantage because the customized isolation of a specific phage is needed for each newly discovered pathogen or each treated patient. Additionally, a bacterium can quickly develop resistance to prevent phage adsorption by blocking or covering its surface receptors via spontaneous mutation or phenotypic variation [22,23]. The prime cause of bacterium resistance to a phage is downregulation of the surface receptor [24,25]. As a result, a phage cocktail, a combination of multiple wild-type phages each targeting a specific cohort of pathogens or mutants, is applied to circumvent the limited host range and the emergence of phage resistance [11,21,26,27]. While phage cocktails have proven successful in some clinical trials [28,29,30], they have labor-, time-, and cost-intensive isolation and production processes and may pose additional regulatory hurdles [12,31]. Therefore, there is increasing interest in engineering a characterized phage (mainly by manipulating the tail fibers or spikes) with a reprogrammed or expanded host range to bypass the need for isolation of new phages and modulation of phage cocktail compositions.

The remarkable diversity of tail fibers makes it challenging to systematically understand the molecular mechanisms of phage recognition and adsorption by its host. The understanding of phage–host interaction will provide the crucial basis and guidance for phage-based therapeutics in medicine [7,10,25,32]. In this review, we comprehensively summarize how the tail fibers of the T4 phage recognize host surface receptors at single-molecule and atomic levels. In addition, we briefly summarize the general strategies of tail fiber engineering to produce antibacterials with a reprogrammed range, including machine learning technologies for predicting phage–host interactions.

## 2. Molecular and Structural Insight of the Interaction between T4 Phage Long Tail Fibers (LTF) and *Escherichia coli* Receptors

### 2.1. T4 Phage Architecture and the Structure of Long Tail Fibers (LTF)

Phage T4, the best-characterized phage studied to date, belongs to the *Straboviridae* family [6] (https://ictv.global/taxonomy (accessed on 1 October 2022)) and infects the Gram-negative *Escherichia coli* (*E. coli*) and closely related *Shigella* species [33]. T4 capsid, a 120 nm long and 86 nm wide prolate icosahedron, encapsidates a ~171 kbp linear, double-stranded DNA genome, and its exterior is decorated with Hoc (highly antigenic outer capsid protein) and Soc (small outer capsid protein) nonessential proteins (Figure 1A), which can be fused with foreign proteins for various biomedical applications [34,35,36,37,38,39]. T4’s circularly permuted genome consists of ~289 protein coding sequences and encodes 40 structural proteins, with most of them involved in tail assembly (Figure 1B) [40,41,42]. The capsid has a unique portal vertex [43] to which a 140 nm long contractile tail is attached via a neck/connector complex [44,45]. The collar and whiskers formed by Wac (or fibritin) are assembled just below the capsid–tail junction [46]. The tail comprises an interior rigid tube, an exterior contractile sheath surrounding the tube, and a hexagonal baseplate at the tip of the tail (Figure 1A) [47,48,49]. Two types of fibers, six 145 nm long tail fibers (LTFs) and six 40 nm short tail fibers (STFs), are attached to the baseplate (Figure 1A) [50,51]. The two sets of tail fibers confer to T4 phage one of the most effective infection efficiencies [52]. The T4 LTFs determine its host specificity via interacting with bacterial surface receptors.

The T4 LTFs can reversibly interact with Outer membrane porin C (OmpC) and LPS receptors exposed on the surface of *E. coli* K12 and *E. coli* B strains, respectively, initiating the adsorption process, which is the first step in the T4 lytic life cycle [8,50,53]. The kinked LTF consists of four structural gene products (gp) (Figure 1C): gp 34 (140 kDa, 1289 amino acids), gp35 (35 kDa, 372 amino acids), gp 36 (23 kDa, 221 amino acids), and gp 37 (109 kDa, 1026 amino acids), with a stoichiometry of gp34/gp35/gp36/gp37 of 3:1:3:3 [54]. The long and thin LTF can be divided into ~70 nm proximal and ~75 nm distal half-fibers (proximal and distal are in relation to the assembled LTF to the tail baseplate), hinged at an angle of around 160° [8,54]. The proximal half-fiber is formed by a homotrimer of gp34, followed by the hinge composed of monomeric gp35, whereas the distal half-fiber is formed by homotrimers of gp36 and gp37 (Figure 1C) [55]. Additionally, LTF shows a somewhat linear arrangement of the four proteins. Thus, the N-terminal end of gp34 binds to the baseplate periphery, while its C-terminal end attaches to gp35. Similarly, the N-terminal end of gp36 attaches to gp35, while the C-terminal end of gp36 binds to the N-terminal end of gp37. Finally, the C-terminal end of gp37 contains the extremely distal receptor-binding domain (RBD or “tip”) responsible for recognizing the host receptors (Figure 1C,D) [50].

The T4-encoded molecular chaperone gp57A is required for the correct trimerization of gp34 and gp37, and another chaperone, gp38, is required for the proper folding and functionality of gp37 [56]. Both gp57A and gp38 are absent in the final assembled T4 virion. For LTF assembly, homo-trimeric gp34 and gp37 assemble independently. Initially, trimeric gp36 proteins assemble on the N-terminal region of the gp37 trimer to form the distal half-fiber, and then monomeric gp35 joins to the gp36 free end to form a gp35–gp36–gp37 complex. The proximal half-fiber gp34 trimer attaches to the gp35–gp36–gp37 complex to form the final complete LTF, which then can coaxially attach to the C terminal domain of gp9 located at the upper edge of the tail baseplate (Figure 1D) [34,50].

No atomic resolution structures of whole LTFs have yet been presented because the large size and simple linear structure lead to poor crystallization [47], although the T4 LTF is one of the best-characterized tail fibers to date [7,57,58]. Seventeen mass domains of variable size and spacing have been observed in the intact LTF by scanning transmission microscopy (Figure 1D) [54]: five domains in the proximal half-fiber gp34 (P1 to P5), a single domain in the gp35 hinge, and eleven domains in the distal half-fiber gp36–gp37 (D1 to D11). Domains D1 and D2 close to the hinge are probably made of gp36, while domains D3 to D11 are formed by gp37 (Figure 1C,D) [51,54].

The atomic structure of trimeric D10 and D11 domains at the C terminus (residues 811–1026 of the 1026 aa gp37) has been determined [57]. This 20 nm-long “needle” region consists of a globular “knob” (~45-Å wide, D10), an elongated “stem” (~15-Å wide, D11), and a small “tip” (~25-Å wide, D11) (Figure 1E). Each chain of the interwoven trimer emanates from the “knob” to the end of the “tip”, twists around a neighboring chain, and turns back, with both the N and C terminus located at the “knob”. The D11 domain (residues 882–1019) inserts into the D10 “knob” domain (residues 811–881 and 1010–1026). D11 consists of two sub-domains: “stem” (residues 882–931 and 960–1009) and “tip” (residues 932–959) (Figure 1E). Most of the D11 amino acids are found in an extended conformation forming the elongated “stem” subdomain. The compact and interwoven “tip” subdomain, inserted into the “stem” subdomain and located at the distal pole of the LTF, plays a primary role in the interaction with host receptors: *E. coli* B type LPS (B-LPS) and *E. coli* K12 type OmpC (K12-OmpC) (Figure 1E) [51,57,58].

### 2.2. Molecular and Structural Insight of T4 LTFs’ Interaction with Host Receptors LPS and OmpC

Bioinformatic analysis suggests extensive sequence conservation of tail fibers from various phages and prophages, except for the “tip” domain [53,57,61]. The LTF “tip” has diverged with distinct shapes and sizes to acquire specific receptor-binding properties [13,14,18,57,62,63,64,65]. Therefore, understanding the molecular mechanism of interaction between the “tip” and host receptor will provide the basis for reprogramming the phage–host interaction. Here, we highlight the molecular mechanisms of T4 LTF “tip” binding to the terminal glucose of B-LPS and to K12-OmpC, and then we summarize the structural model of LTFs during T4 infection initiation.

#### 2.2.1. T4 LTF “Tip” Binding to LPS

LPS, a large glycolipid, is abundant in the outer membrane of Gram-negative bacteria (around a million molecules per cell) and is the primary receptor for phages [66]. LPS is generally composed of three structural domains: lipid A, the oligosaccharide core, and the distal polysaccharide (or O-antigen) [66]. Lipid A is hydrophobic and forms the outer leaflet of the bacterial outer membrane. The core oligosaccharide is a non-repeating oligosaccharide that is divided into two linked moieties: the inner core and the outer core [67]. The inner core is bound to the extracellular side of lipid A, while the outer core is extended from the inner core. The core usually contains glucose, galactose, heptose, and 3-deoxy-D-manno-oct-2-ulosonic acid (Kdo), which can be further modified with phosphates, N-acetylglucosamin, and other substituents [67,68]. The O-antigen is attached to the outer core and consists of a repeating oligosaccharide (two to eight sugars). The LPS is classified into one of two varieties, smooth LPS (e.g., *E. coli* O157) or rough LPS (e.g., *E. coli* B and K12), based on the presence or absence of O-antigen, respectively [69]. The LPS can also be divided into five types based on the constitution of the oligosaccharide core [70,71], such as B-LPS and K12-LPS, which have different outer cores (Figure 2A).

Phages can recognize the core or O-antigen with their tail fibers. Generally, O-antigens differ highly between different bacteria, while the core is more conserved [72]. Thus, phages targeting the O-antigen (smooth LPS) have a narrower host range compared to those specific to the core (rough LPS). It is well characterized that rough LPS of *Shigella* and *Escherichia* is the receptor for T-phages, specifically T3, T4, and T7 [72]. Further, O-antigen is demonstrated to strongly inhibit T4 phage adsorption [68]. The T4 phage recognizes B-LPS of *E. coli* B strain having two terminal glucose (Glu) residues (Glu I and Glu II) in the outer core, but it does not recognize the K12- or O157-LPS having additional sugar residues or branches (Figure 2A) [68,73,74,75]. Interestingly, when the K12- or O157-LPS is mutated to have terminal glucose in the outer core, the T4 phage can bind to these mutants [68,76]. These studies indicate that the T4 LTF “tip” domain might interact with terminal Glu I or with both Glu I and Glu II in the LPS outer core.

How does the T4 LTF “tip” region interact with the terminal glucose of LPS? Generally, protein–saccharide interactions involve the stacking of sugar residues onto side chains of aromatic amino acids, including phenylalanine (F), tryptophan (W), and tyrosine (Y) [77]. Additionally, amino acids with positively charged side chains, including arginine (R) and lysine (K), might also interact with the phosphate groups of the LPS [57]. Therefore, in the “tip” region (residues 932–959), Y932, W936, K945, Y949, Y953, and R954 are potential candidates for direct binding to the bacterial LPS receptor. Furthermore, some studies have utilized genetic and/or biochemical approaches to provide a more in-depth insight into the critical amino acid residues involved in LPS interaction (Table 1) [33,57,58,68,75]. Unexpectedly, a larger percent of residues (16 out of a total of 28) at the “tip” region are found to be involved in LPS receptor interaction, including I933, N937, G938, T939, G940, G942, G943, K945, M946, S947, Y949, I951, S952, Y953, R954, and A955 (Table 1) (Figure 2B) [33,57,58,68,75]. They are divided into two types as follows:

(1) “Loss-of-function” residues (no infection to B strain): When these residues are mutated, the produced LTFs cannot interact with the LPS of the *E. coli* B strain, leading to aborted T4 phage infection (no plaque formation). These individual or combined mutations include I933A; N937A; G938A; T939I; G940A; G942A; G943A; M946V and S947P; G940V, M946V, and S947P; S947A; and Y949A. At the bottom-most portion of the “tip”, there are three small cavities. Each cavity volume is suitable for accommodation of one glucose moiety, suggesting that this cavity might be the binding site for the terminal glucose of the LPS receptor (Figure 2B) [57,58]. Residues G940, G942, G943, S947, and Y949, forming a bottom patch, occupy part of the cavity and face the outer core of LPS when the T4 phage lands on the surface of the *E. coli* B strain. Glycine has a wider phi/psi angle (ψ = 180° and ψ = 0°), likely for maintaining the whole structure or conformation. Enriched glycine residues (G938, G940, G942, and G943) are observed in the “tip” region of the T4 phage as well as in other phages such as T2 and S16 [17,78], indicating the important role of glycine in receptor interaction. Furthermore, in the bottom patch, the Y949 aromatic side chain protrudes out at the bottom rim of the cavity and may directly stack onto the LPS terminal glucose for binding. Additionally, residues I933, N937, G938, and T939 are part of a hydrophobic β-sheet core, and I933 is linked to N937, G938, and T939 residues, which are located at the upper rim of the cavity. Notably, T939 is flanked by glycines, allowing the side chain of threonine to protrude out, probably for interaction with LPS as well (Figure 2B).

(2) “Host-range-expanded/shifted” residue: When this residue is mutated, the produced LTFs can interact with other types of LPS that lack the terminal glucose, such as *E. coli* K12 LPS [58,68,75] or *Yersinia pestis* LPS [33,79], leading to infection by the T4 phage to the expanded or shifted hosts. The studied mutations include G938V; K945A; I951A; S952K; Y953A; Y953H; Y953R; and R954S and A955T. Interestingly, compared with most “loss-of-function” residues located at the bottom of the “tip”, most “host-range-expanded/shifted” residues are located at the lateral side of the “tip”. The mutations K945A, I951A, S952K, Y953A, Y953H, and Y953R lead to the LTFs interacting with not only *E. coli* B LPS but also *E. coli* K12 LPS (expanded). Additionally, the mutations G938V or R954S and A955T lead to the loss of *E. coli* B LPS interaction but likely gain binding capacity to *Yersinia pestis* LPS, which lacks the outer core and is composed of lipid A and inner core (shifted). Notably, Y953 is located at the upper rim of the cavity (or the border between the bottom and lateral surface) and is another aromatic residue that has the potential to stack its aromatic chain onto the sugar moiety of LPS. When this Y953 is mutated to another residue (A, H, or R), the LTF binding range is expanded, indicating that Y953 might directly determine LPS binding specificity. Additionally, K945 and R954, with a positively charged side chain, have the potential to interact with the phosphate groups of LPS and might also directly determine LPS binding specificity (Figure 2B).

Collectively, it is surprising to see so many amino acid residues at the “tip” involved in the LPS interaction. These residues are selected by nature and appear to form diverse patches randomly scattered on the bottom and the lateral side of the LTF trimeric “tip”. Structural docking analysis shows that the patches on the bottom can accommodate the adsorption of LPS by the terminal glucose [58]. In addition, the patches on the lateral side play a critical role in the LPS binding and specificity determination. It seems that the “tip” binds to the sugar moieties of the LPS in several locations along its length [53]. These diverse patches likely contribute to efficient binding between T4 phage LTFs and bacterial LPS receptors in the initial adsorption step. However, the interaction between the “tip” and the LPS is weak and unstable. A binding-strength study using atomic force microscopy [80] showed that the interaction between T4 gp37 and LPS from *E. coli* B (host) was quantified as 70 ± 29 pN, while interaction between T4 gp37 and non-host LPS was measured as 46 ± 13 pN (negative control). Such efficient but weak/unstable interaction would lead to a dynamic “association–dissociation” equilibrium (dynamic and irreversible), endowing the T4 phage with the capability to randomly walk across the bacterial surface to search for an optimal site [58]. Furthermore, this irreversible interaction probably orients and/or fixes the LTF perpendicular to the cell surface for efficient infection [13,14,51].

#### 2.2.2. T4 LTF “Tip” Binding to OmpC

Porin serves as an aqueous pore that is abundant in the outer membrane of Gram-negative bacteria and facilitates the nonspecific diffusion of nutrients and water-soluble drugs, with a molecular mass cut-off of about 600 Da [24,82]. In *E. coli*, two major porins, OmpC and OmpF, represent more than 50% of the total protein integrated into the outer membrane [24]. OmpC is the primary receptor for the T4 phage to infect the *E. coli* K12 strain because T4 cannot adsorb to a K12 mutant lacking OmpC [68]. It is known that OmpC also serves as a receptor for other phages, such as Tulb, Hy2, AR1, and ss4 [72,83]. OmpC is organized as a trimer. Each monomer shows a β-barrel structure formed out of 16-stranded antiparallel β-sheets, with 8 internal periplasmic turns and 8 extracellular loops that connect each β-sheet [82]. Structural docking analysis has shown that the size of the LTF “tip” domain (~25 Å) is similar to the size of the surface cavity formed by the trimeric OmpC molecules [57]. The “tip” fits snugly into the OmpC outer cavity and most likely interacts with the extracellular loops. The extracellular loops 1, 4, and 5 in K12-OmpC are required for efficient T4 phage adsorption (Figure 2C) [75].

A number of residues at the “tip” are also found to be involved in binding to K12-OmpC. The residues that have been studied to date are: I933, N937, G938, G940, V941, G942, G943, K945, M946V, S947, Y949, I951, Y953, and A955 (Table 1) (Figure 2B) [33,57,58,68,75]. Interestingly, almost all these residues (except V941) overlap with the residues involved in B-LPS binding. Similarly, these residues are divided into two types: (1) “Loss-of-function” residues and (2) “host-range-expanded/shifted” residues. For “loss-of-function” residues, some mutations, including G940A, V941E, G942A, G943A, G943S, K945A, S947A, Y949A, I951A, Y953A, Y953R, and A955E, result in loss of interaction between the LTFs and K12-OmpC. Most of these mutations also lose infection of the *E. coli* K12 strain, except residues K945A, I951A, Y953A, and Y953R, which obtain the capacity of binding to K12-LPS as compensation for infection. As indicated by structural docking analysis, residues G940, G942, G943, S947, and Y949, lining the cavity surface of the “tip” bottom, interact with the amino acid residues exposed in the barrel cavity of the K12-OmpC receptor, probably via a combination of hydrogen bonds, hydrophobic interactions, and shape-complementary van der Waals interactions [58]. In addition, residues K945, I951, and Y953, located above the upper rim of the bottom cavity, are also suitable for interacting with residues lining the K12-OmpC barrel cavity [58,75]. 

For “host-range-expanded/shifted” residues, the combined mutations G940V, M946V, and S947P; or M946V and S947P in the “tip” result in the loss of K12-OmpC interaction but gain binding capacity to *E. coli* O157 OmpC (shifted) [75]. The LTF with mutation N937S shows reduced binding to wild-type K12-OmpC but gains binding capacity to K12-OmpC-P177V mutant (expanded). Additionally, the LTF with mutation G942R loses the interaction with wild-type K12-OmpC but obtains binding capacity to K12-OmpC-F182A mutant (shifted). Residues P177 and F182 in loop 4 of K12-OmpC, aligned vertically and exposed toward the central part of the OmpC barrel, are the key residues for LTF interaction. The wild-type T4 phage is not able to infect the K12 strain with OmpC P177V or F182A mutation (Figure 2B) [75].

Although receptors LPS (sugar) and OmpC (protein) are structurally completely different, the critical amino acid residues in the “tip” involved in the interaction with them are quite similar. The diverse patches formed on the bottom and lateral sides of the trimeric “tip” function for both B-LPS and K12-OmpC receptor interaction. The LTF “tip” binds to the amino acid residues exposed in the barrel cavity of OmpC in several locations along its length (Figure 2D). Similar to the B-LPS interaction, the K12-OmpC interaction is also efficient but weak/unstable (dynamic and irreversible), facilitating the random moving of T4 phage on the bacterium surface to search for an optimal site for infection: there are 10^5^ copies of OmpC on the bacterial surface, but T4 prefers infection sites at cell poles or at an invagination of an upcoming division site [84,85]. Interestingly, a T4 phage with a “tip” mutation including I933A, N937A, or G938A loses infectability of the K12 strain, though these mutants exhibit a 2–3-fold increase of K12-OmpC binding compared with the wild-type [58]. This tight binding might prevent any subsequent movement of T4 on the surface. These mutations might compromise the dynamic and irreversible interactions of LTFs with host OmpC receptors.

Notably, mutation K945A, I951A, Y953A, or Y953R loses binding to the original K12-OmpC receptor but obtains binding capacity to a new sugar receptor K12-LPS, which provides a significant clue for more-effective phage therapeutics in the future. The reason is that the porin or Omp protein, allowing passage of drugs such as antibiotics, might not be the optimal phage receptor in practical phage therapy. If used in therapy, phage-resistant bacteria with downregulated or even defective porin might be selected. Then it would be more difficult and terrible to use antibiotics to treat these mutated bacteria because the drug passage channels will be diminished or may have vanished altogether [22,24,86,87].

### 2.3. Model of Long Tail Fibers (LTFs) during T4 Infection Initiation

Since phages do not have specific motion structures to move independently, initial adsorption results from random phage–host collision described by the Law of Mass Action. T4 phage infection is initiated with host recognition in which the LTF “tip” specifically and reversibly recognizes the LPS or OmpC receptor on the cell wall. The *E. coli* cell wall consists of two concentric lipid bilayers, the outer membrane and the inner (cytoplasmic) membrane, with peptidoglycan periplasm between [88]. Upon receptor recognition at the suitable site on the cell wall, a mechanical signal is transferred to the phage baseplate, causing conformation change of the baseplate. The baseplate-anchored short tail fibers (STFs) are then unpinned, rotate downward, and irreversibly bind to the lipid A-inner core region of LPS. The baseplate completes the conformation conversion from hexagonal to star shape during the tail-fiber-binding process and is oriented parallel to the cell surface. Then, contraction of the tail sheath is trigged, pushing the hollow tail tube through the host outer membrane and periplasm. The inner membrane bulges from its normal plane to fuse with the phage ejection nanomachine. Finally, a channel across the outer and inner membranes is formed, facilitating the injection of phage genome DNA into the bacterial cytoplasm for the synthesis of new virions. The infection details have been reviewed elsewhere [7,8,49,50,89]. Here, we highlight the model of LTFs during recognition and infection initiation, which plays an important role in determining the host range (Figure 3).

In the uninfected state, the six LTFs exhibit two different conformations: extended and retracted [90]. The extended and mobile LTFs extend away from the baseplate and are free to contact the bacterial surface, while the retracted LTFs are wrapped around and folded upwards against the tail sheath, wac (whisker and collar), and capsid. The structural reconstruction [51] shows that the proximal half-fiber (gp34) is folded back and wrapped around the tail sheath. The hinge domain (gp35) binds to the tip of the whisker, the D7 domain (gp37) of distal half-fiber to the collar, and the D10–D11 domains (gp37) to the capsid. The key amino acids mediating these interactions are yet unknown. Interestingly, in freshly purified phages, not all fibers are retracted or extended at any given time. Most phages have three to four retracted LTFs on average. Maintaining major LTFs as retracted not only confers some stability against premature baseplate conversion and tail sheath contraction before infection, but also reduces dimensionality to allow faster phage diffusion, reversible adsorption, and walking across the host surface for a suitable infection site. Additionally, each individual fiber seems to be in a dynamic “retracted–extended” equilibrium that does not need chemical energy to maintain (Figure 3).

Since there are a number of receptor-binding patches on the T4 LTF “tip” and abundant LPS or OmpC receptors on the host surface, a collision between the “tip” and the receptor should be relatively frequent. The first specific recognition involves a single extended LTF bound to its receptor LPS or OmpC. The initial temporary interaction prevents phage diffusion and hinders the bound tail fiber from returning to be retracted [51]. Additionally, the trimeric “tip”, having symmetrical and diverse binding patches, allows weak and unstable interaction with the receptor in several locations along its length. As a result, the “tip” might move up and down as well as rotationally like a molecular pivot [58]. Then, a second extended LTF binds to another receptor on the surface before the first one has dissociated and/or retracted. Both “association–dissociation” of receptor–LTF and “extended–retracted conformation” of LTF is likely in a dynamic equilibrium. Repeating these dynamic processes allows the T4 phage to randomly walk across the bacterial surface to search for an optimal site before committing to infection. These continuous “touch and search” efforts of the phage finally lead to the discovery of the optimal site for efficient infection, such as a cell pole or an invagination of an upcoming division site for T4 phage (Figure 3) [84,85].

## 3. Engineering Strategies of Phage Tail Fiber for Reprogramming Phage Host Range

The interaction between a bacteriophage and its host is mediated by the phage’s tail fiber “tip” domain or receptor binding domain, which is thus the main engineering site for reprogramming the phage host range. In earlier days, researchers relied on the natural evolution process to increase the host range of phages. Then, the identification and characterization of the phage receptor binding domain at molecular/atomic levels allow the engineering of the tail fiber to reprogram the phage host range. More recently, the advancement of bioinformatics, machine learning, and artificial intelligence could allow rapid identification of phage–host interaction based on the characterized genome sequences of various phages and bacteria.

### 3.1. Host Range Widening through Natural Evolution

There is an evolutionary arms race between bacteria and phages [91]. Bacterial hosts have evolved multiple anti-phage tactics, such as phage receptor blocking and the CRISPR-Cas system [22,92]. On the other side, phages have also evolved corresponding strategies to avoid or circumvent these selecting pressures, such as tail fiber mutation to recognize new receptors and the anti-CRISPR system [93,94,95]. Researchers have been practicing this natural evolution strategy to extend the phage host range. This strategy involves growing wild-type phages in various hosts for several generations to generate mutants (mainly in tail fibers) that infect new hosts. For example, the host range of T7 phage was initially limited to *E. coli* and a few *Shigella* strains. Using natural evolution, the host range was extended to *Yersinia pestis* [96]. In another approach, Burrows et al. applied a phage cocktail to phage-resistant and -sensitive bacteria. After 30 rounds of selection, recombinant phages were isolated with significantly extended host ranges [97]. The limitation of natural evolution is its requirement for extensive co-culture and a labor/time-intensive selection process.

### 3.2. Rational Genetic Engineering of Tail Fibers

The rational genetic engineering approach requires an extensive molecular understanding of phage tail fibers that interact with hosts. The phage host range can be reprogrammed via swapping the tail or tail fiber genes with those from other phages by homology-directed recombination or synthetic engineering. Yoichi et al. changed the host range of T2 phage by swapping its *gp37* and *gp38* (two gene products at the tip of T2 long tail fiber) with the corresponding gene products from phage PP01 to produce a recombinant T2 phage. T2 phage normally infects *E. coli* K12, whereas PPO1 infects *E. coli* O157: H7. Interestingly, the recombined T2 phage could no longer infect its native host, *E. coli* K12, but was able to infect *E. coli* O157: H7 [98]. Additionally, Ando et al. developed a synthetic biology strategy for extending the host range of T7 phage via swapping whole-tail components (including tail fiber) [99]. They used a yeast-based platform for phage genome engineering, then they transformed the modified phage genomes into *E. coli* to reboot phages with novel host ranges. Using this technology, they diverted *E. coli* T7 phage to efficiently target and kill new bacterial hosts, including *Yersinia* and *Klebsiella* [99].

In addition, the phage host range can also be reprogrammed by random mutation or directed evolution in the phage tail or tail fiber region. Yosef et al. developed GOTrap (General Optimization of Transducing particle) technology to engineer T7 phage for extending foreign DNA transduction into new bacterial hosts (Figure 4A) [100]. In this study, the T7 wild-type whole-tail genes (*gp11*, *gp12*, and *gp17*) were deleted from the phage genome. Then, T7 phages lacking their tail genes were applied to infect *E. coli* hosts encoding randomly mutated tails (by chemical mutagenesis) in a packable plasmid with a selectable marker, resulting in numerous variants of tail-mutated phage particles. As a result, the selected phages were characterized and able to deliver the desired DNA to broad new bacterial species, including *Klebsiella*, *Shigella*, *Salmonella*, *Escherichia*, and *Enterobacter* (Figure 4A) [100]. The major limitation of this technology is the generation of a large plasmid library (~10^12^ to 10^15^) for creating multiple simultaneous mutations in the tail genes [100].

For widening the host range of T3 phage (a T7-like phage), Yehl et al. adopted structure-informed engineering of viral tail fibers to generate host range alterations [101]. Using homology modeling with the structurally characterized T7 phage, they generated the tail fiber structure of T3 and identified similar distal loops located at the tail fiber tip. These loops act as the host-determining region that binds to the LPS receptor of the host bacterial strain. Inspired by antibody specificity engineering, they created a T3 tail-mutated library with high-throughput mutations in the four outward loop regions containing ten million different members (Figure 4B), which was named “Phagebody”. The engineered phage showed increased or altered host range and also could efficiently reduce the growth of both antibiotic- and phage-resistant bacterial strains in in vitro and in vivo mouse models [101]. As more and more viral tail and tail fiber components are structurally resolved and characterized, the breadth of viral models to which this “Phagebody” strategy can be used will also be extended.

As we summarized in T4 LTF “tip” and receptor interaction, the majority of amino acid building blocks of the “tip” region are critical for bacterial receptor interaction. Likewise, in T7 phage, Huss et al. [102] demonstrated that the building blocks of the T7 receptor binding domain are important for efficiency and specificity. Even small changes to the receptor binding domain can make a big difference to the T7 phage’s ability to infect its hosts. They designed an ORACLE method (Optimized Recombination Accumulation and Library Expression) to create a T7 phage mutation library containing all single amino acid substitution (1660 variants) in the tip domain (Figure 4C). The role of each building block in the T7 receptor binding domain was meticulously dissected, and hundreds of function-enhancing substitutions were identified [102]. These studies help to understand exactly how the tail fiber allows a virus to infect a specific type of bacteria and could pave the way for fighting increasingly resistant bacterial infections.

### 3.3. Bioinformatic Prediction of Phage Host Range

The phage host range can be evaluated or predicted using a bioinformatic approach. Some phages and their hosts share common evolutionary ancestry; in other words, there is sequence homology between these phages and their hosts. As a result, it is easy to identify the host range of phages using common alignment tools such as BLAST (Basic Local Alignment Search Tool) [61]. If there is no significant sequence similarity between the phage and its host, other genomic features, such as codon usage, sequence composition, oligonucleotide frequency, and k-mer composition (k-mer is used to analyze nucleotide composition) can also be used to identify the host range. There are some tools available online to identify these features, such as VirHostMatcher, which identifies host range based on oligonucleotide frequency in a k-mer length [103]. Lastly, the advancement of high-throughput sequencing technology allows prediction of host range based on sequence features, CpG bias, and CG bias through machine learning algorithms [104,105].

In the machine learning approach, metagenomic datasets can be directly used to identify the phage receptor binding protein (tail fiber); in addition, it allows the identification of new receptor binding proteins [106]. Furthermore, using the metagenomic dataset, a targeted phage with an extended host range can be generated by swapping the tail or tail fiber [107]. Boeckaerts et al. developed a machine learning tool for predicting phage host specificity based on the annotated receptor binding protein sequence data [108]. From the raw DNA and RNA sequences, they generated 218 features, from which nucleotide frequency, TTA codon frequency, TTA codon usage bias, first Z scale descriptor (lipophilicity of amino acids), and GC content scored among the top five features for having higher influence on the prediction model. They evaluated the predictive performance of the model by using four machine learning methods. When they compared the final model with BLASTp, they reported the final model outperformed BLASTp when the sequencing similarity among other known sequences in the database was less than 75% [108] (Figure 5A).

To systemically engineer the phage genome, deep learning (a subset of machine learning) can also be used. Ataee et al. proposed a two-component deep learning model: the first component is the PERFHECT model that predicts the interaction between bacteria and phages using a 1-D convolutional neural network (CNN); and the second component is the PERPHECT generator that alters the existing phage genome to enhance host range prediction (Figure 5B) [109]. In the predictor model, they used genomic information from both phages and bacteria to predict interactions. The predictor used 46 strains of *Pseudomonas aeruginosa* to predict the host range, and the generator model modifies 42 phages’ genomic sequences for precise host range engineering. The generator model’s training accuracy level is reasonably high (~96%), and it can enhance the host range of 18 out of 42 phages. This deep learning method allows systemic modification of the phage genome for host range engineering, which generates superior phage variants that overcome limitations due to bacterial resistance against natural phages.

## 4. Conclusions

The interaction between phage tail fiber and bacterial surface receptor determines the phage host range. Even a single amino acid change to the fiber “tip” domain can make a big difference to the phage’s ability (activity and specificity) to infect its hosts. The exact mechanisms of how the tail fiber interacts with the receptor at the molecular/atomic level are critical for engineering phages with reprogrammed host ranges. The advancement of technologies such as gene editing and bioinformatic analysis is greatly accelerating the characterization and engineering of phage–host interaction. Engineered phages with reprogrammed host ranges would bypass the need for isolation of new phages and modulation of phage cocktail compositions and would pave the way for more-effective phage therapeutics in the future for the increasing and serious challenge of antimicrobial resistance.

## Figures and Tables

**Figure 1 ijms-23-12146-f001:**
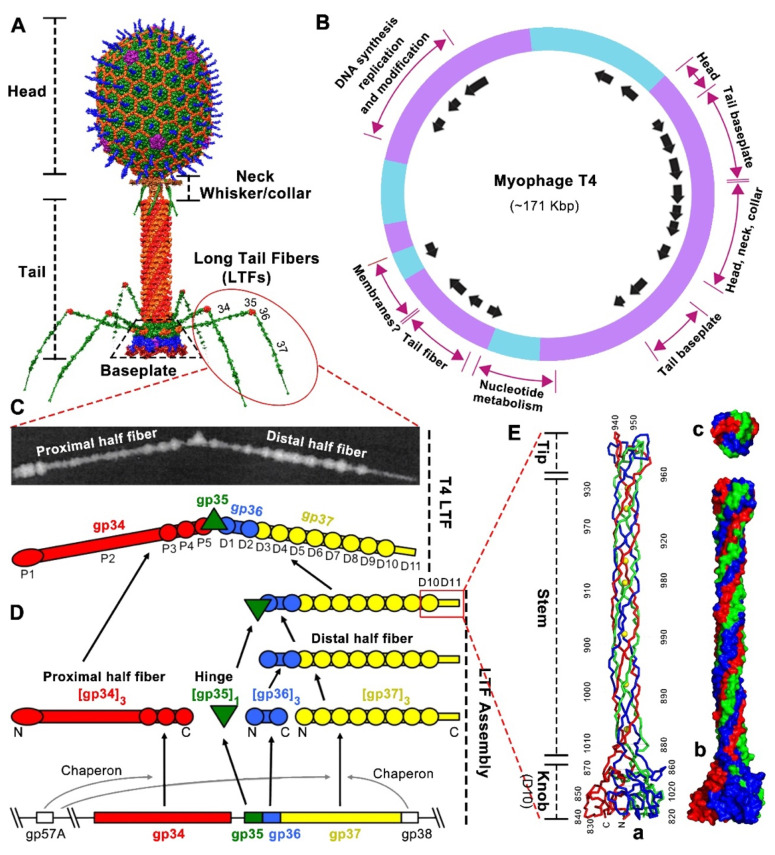
T4 phage architecture and the structure and assembly of long tail fibers. (**A**) T4 phage architecture [59]. (**B**) T4 genetic map showing gene clusters with related functions and origin and direction of transcripts (arrows) [60]. (**C**) The structure and schematic of T4 long tail fiber with seventeen mass domains observed by scanning transmission microscopy [54]. Reprinted with permission from Elsevier. (**D**) The assembly of T4 long tail fiber [8]. (**E**) The structure of the T4 long tail fiber “needle” (D10 and D11) responsible for host receptor recognition [57]. (a) Ribbon structure of the trimeric “needle” consisting of knob, stem, and tip. The N and C termini and every 10th residue of one chain are labeled. (b,c) Surface structure of the trimeric “needle” seen from the side (b) and top (c).

**Figure 2 ijms-23-12146-f002:**
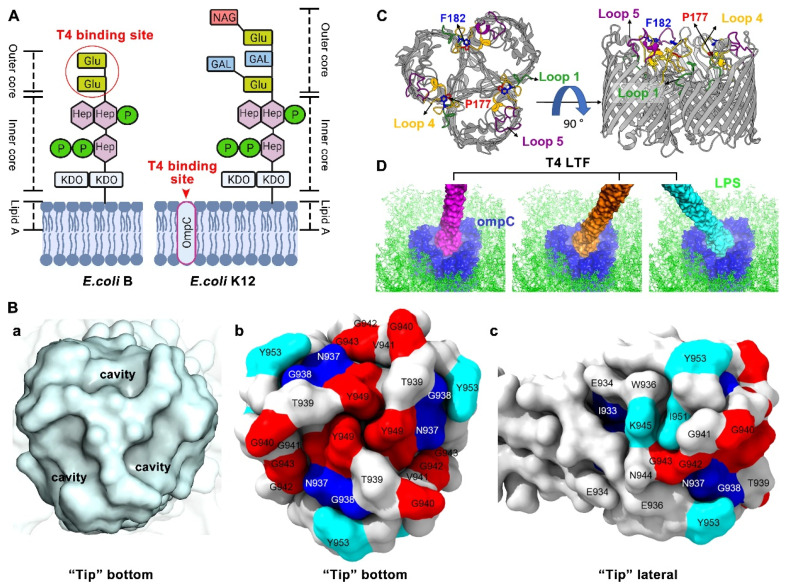
Molecular and structural insight of T4 long tail fiber interaction with host receptors LPS and OmpC. (**A**) Structural schematics of T4 LPS receptors on the surface of *E. coli* B and K12 strains: Glu, Glucose; Hep, L-glycero-D-manno heptose; P, Phosphate; KDO, 3-deoxy-D-manno-oct-2-ulosonic acid; GAL, Galactose; and NAG, N- acetylglucosamin. Created by BioRender.com. (**B**) The “tip” surface structures and the critical amino acid residues involved in host receptor binding [58]. (a) The bottom surface structure of the “tip” showing three small cavities, each suitable for the accommodation of one glucose moiety. (b) The key residues on the bottom surface of the “tip”. (c) The key residues on the lateral surface of the “tip”. (**C**) OmpC viewed from the top (left) and the side (right) [75]. The T4 phage LTF binding components, loops 1, 4, and 5, and residues P177 and F182, are highlighted. The outer and inner membranes of *E. coli* K12 are indicated by the gray bars. (**D**) The T4 LTF “tip”-OmpC docking model at different angles [58]. Illustrations 2C and 2D were re-made by UCSF ChimeraX [81].

**Figure 3 ijms-23-12146-f003:**
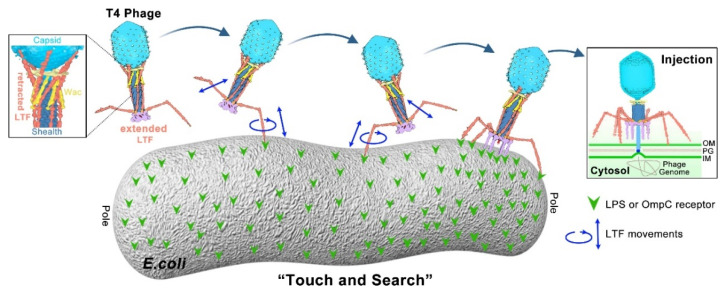
“Touch and Search” model of LTFs during T4 infection initiation. In the free state, most phages have three to four retracted LTFs on average. Each fiber is likely in a dynamic “retracted–extended” equilibrium that does not need chemical energy to maintain. Upon infection, the trimeric LTF “tip” allows weak and unstable interaction with the receptor, probably causing the “tip” to move up and down as well as rotationally (“association–dissociation” equilibrium). The “association–dissociation” and “extended–retracted conformation” dynamic equilibriums allow the T4 phage to randomly walk across the host surface to search for an optimal infection site [51].

**Figure 4 ijms-23-12146-f004:**
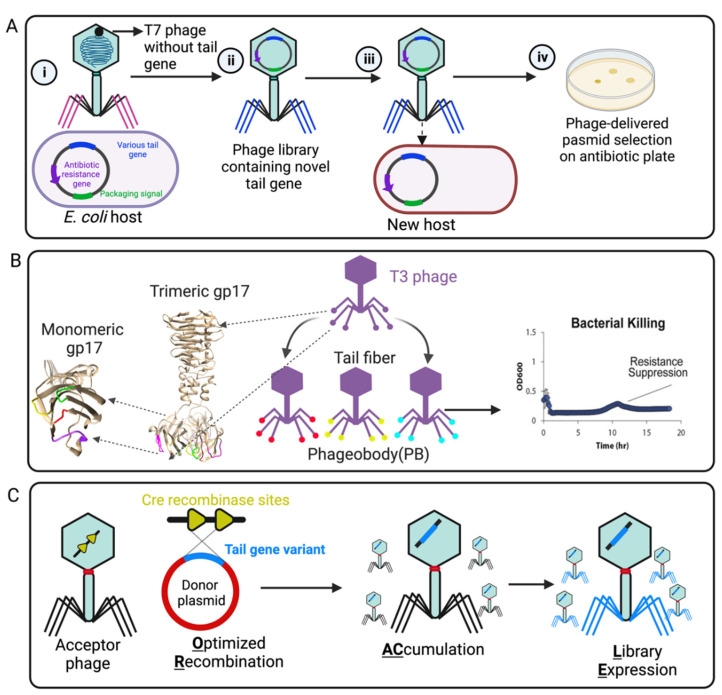
Representative engineering strategies of phage tail fiber. (**A**) GoTrap model [100]: (i) T7 phage without a tail fiber gene infects *E. coli* carrying a plasmid with antibiotic resistance marker, a T7 packaging signal, and various tail components. (ii) The new phage contains a different tail fiber (iii) that can recognize a new host for plasmid transduction. (iv) A new host containing the phage-transduced plasmid is selected on an antibiotic plate. (**B**) Phagebody model: The loop region (tip) of T3 phage gp17 for random mutagenesis is marked by magenta, green, yellow, and red. The selected phageobody can suppress bacterial growth [101]. (**C**) ORACLE method: An acceptor phage is generated in which the tail fiber gene is replaced with a fixed sequence flanked by CRE recombinase sites (a landing site for inserting tail variants) (yellow). The phage variants are then generated within the host by Cre-mediated optimized recombination by inserting tail fiber variants from a donor plasmid (blue, containing single amino acid substitution) into the landing sites [102]. Created by BioRender.com.

**Figure 5 ijms-23-12146-f005:**
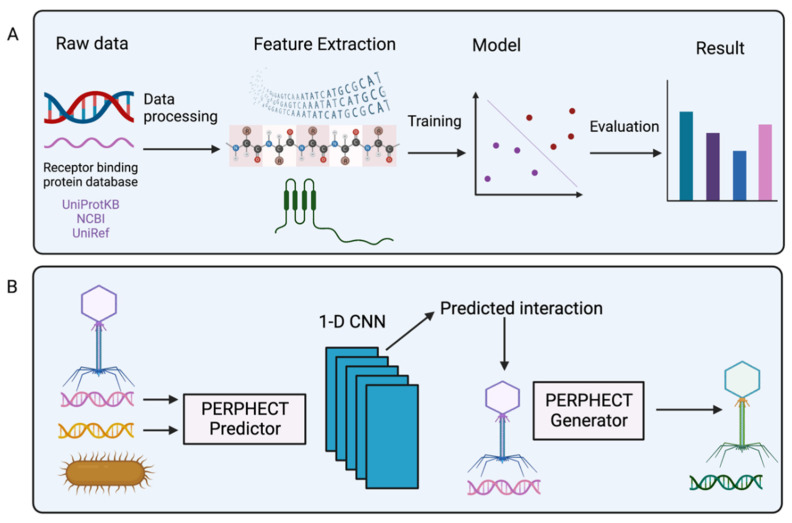
Schematic representation of the machine learning model. (**A**) The raw data (DNA and RNA sequences) of the receptor binding protein database are used to extract features. Next, the features are fitted into different machine learning models, which are evaluated to predict the best result. (**B**) Representation of PERPHECT model; phage and bacterial genetic information are used by the PERPHECT model and PERPHECT generator to provide guidance for genomic modification of the existing phage [109]. Created by BioRender.com.

**Table 1 ijms-23-12146-t001:** The key amino acid residues in the T4 LTF “tip” subdomain involved in the interaction with LPS and OmpC receptors. (*) the binding between OmpC and LTF was determined by in vitro enzyme-linked immunosorbent assay (ELISA) using purified OmpC protein and the LTF needle (gp37; amino acids 799–1026) [58]. (**) the wild-type T4 phage is unable to infect the *E. coli* K12 strain with P177V or F182A mutation in the K12-OmpC receptor. ND, not determined.

Amino Acid Mutations at the “Tip” (932–959 aa)	LPS Binding	OmpC Binding
I933A N937A G938A	No B-LPS binding (No infection to B)	2–3-fold increase of K12-OmpC binding * (No infection to K12)
N937S	ND	Reduced K12-OmpC binding K12-OmpC (P177V) ** binding
G938V R954S and A955T	No B-LPS binding (No infection to B) *Yersinia* LPS binding (Infection to *Yersinia*)	ND
T939I	No B-LPS binding. (No infection to B)	Normal K12-OmpC binding (Infection to K12)
G940A G942A G943A S947A Y949A	No B-LPS binding (No infection to B)	No K12-OmpC binding * (No infection to K12)
G940V and M946V and S947P M946V and S947P	No B-LPS binding (No infection to B)	No K12-OmpC binding (No infection to K12) O157-OmpC binding
V941E G943S A955E	Normal B-LPS binding (Infection to B)	No K12-OmpC binding (No infection to K12)
G942R	ND	Normal K12-OmpC binding K12-OmpC (F182A) ** binding
K945A I951A Y953A	B-LPS binding and K12-LPS binding (Infection to B)	No K12-OmpC binding * (Infection to K12)
S952K Y953H	B-LPS binding and K12-LPS binding (Infection to B)	ND
Y953R	B-LPS binding and K12-LPS binding (Infection to B)	No K12-OmpC binding (Infection to K12)

## Data Availability

Not applicable.

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
