# Peer review of "Understanding Bacteriophage Tail Fiber Interaction with Host Surface Receptor: The Key “Blueprint” for Reprogramming Phage Host Range"

_ijms, 2022, doi:10.3390/ijms232012146_

Round 1

Reviewer 1 Report

Bacteriophages make promising agents for biocontrol of bacterial diseases, but also demonstrate some challenges and limitations. Mourosi et al made a detailed and comprehensive description of the adsorption apparatus of model phage T4. The authors thoroughly describe the molecular mechanisms of T4 tail fiber binding to bacterial receptors, the structure of bacterial receptors and phage RBPs. Also, the manuscript touches importants tasks of engineering strategies of phage tail fiber for reprogramming phage host range and describes the bioinformatic methods of phage host range prediction based on the machine learning approach. Overall, the manuscript is well-written and provides quality illustrations.

The only minor concern is related to the taxonomy description. The Myoviridae family has been abolished recently. Currently phage T4 is classified as a member of Straboviridae family, Tevenvirinae subfamily, Tequatrovirus genus (https://ictv.global/taxonomy).

Line 189 - replace "?" with "."

Reviewer 2 Report

The manuscript by Mourosi et al., is an interesting and detailed review of phage T4 long tail fiber (LTF) tip interaction with T4 receptors at a bacterial cell surface to use this knowledge and the known methods of host specificity modification of other phages to explore the possibility to modify the host range of T4 by targeted mutagenesis of T4 LTF tip. The manuscript is well written and up to date, and summarizes the results of old as well as recent findings concerning the the T4 LTF tip interaction with T4 receptors. It should be of interest for readers of IJMS. I have only some minor comments that are listed below:

L. 33: Replace "order Caudovirales" with "class Caudoviricetes"; see the newest ICTV taxonomy of viruses in Recent changes to virus taxonomy ratified by the International Committee on Taxonomy of Viruses (2022). Walker et al., 2022. Arch Virol. doi: 10.1007/s00705-022-05516-5.

L. 34-35: Replace "into three families Myoviridae, Siphoviridae, and Podoviridae" with " "into three morphotypes myovirus, siphovirus and podovirus". Recent taxonomy changes excluded several families from the former Myoviridae, Siphoviridae and Podoviridae based on the results of comparative and phylogenetic analysis of phage genomes and proteoms. The older morphology-based phage classification has been changed into phylogenetic classification (see ICTV taxonomy at https://ictv.global/taxonomy).

L. 60. Please check again that reference [23] is appropriate.

L. 86: Replace "open reading frames" with "protein coding sequences"

Figure 1D is composed of parts that are direct copies of Figure 15 from Leiman et al., 2010. Morphogenesis of the T4 tail and tail fibers. Virology Journal, 7: 355, doi: 10.1186/1743-422X-7-355. Although the authors provide a citation of Leiman et al. paper, in my opinion this figure requires a copyright permission.

L. 261: Species names should be italicized.

L. 349-352: What about the cell wall, which should be mentioned here for clarity?
